# Identification of the Wheat (*Triticum* *aestivum*) *IQD* Gene Family and an Expression Analysis of Candidate Genes Associated with Seed Dormancy and Germination

**DOI:** 10.3390/ijms23084093

**Published:** 2022-04-07

**Authors:** Mingli Liu, Zhuofan Wang, Chenchen Wang, Xu Pan, Wei Gao, Shengnan Yan, Jiajia Cao, Jie Lu, Cheng Chang, Chuanxi Ma, Haiping Zhang

**Affiliations:** State Key Laboratory of Wheat Biology and Genetic Improvement on Southern Yellow & Huai River Valley, Ministry of Agriculture and Rural Affairs, School of Agronomy, Anhui Agricultural University, Hefei 230036, China; 18256980479@163.com (M.L.); fxf629925@163.com (Z.W.); chen220718@163.com (C.W.); pxbrhhys@163.com (X.P.); hellogw116@163.com (W.G.); 18756059985@163.com (S.Y.); wheatcaojia@163.com (J.C.); lujie@ahau.edu.cn (J.L.); changtgw@126.com (C.C.)

**Keywords:** expression analysis, *IQD* genes, seed dormancy and germination, wheat

## Abstract

The IQ67 Domain (*IQD*) gene family plays important roles in plant developmental processes and stress responses. Although IQDs have been characterized in model plants, little is known about their functions in wheat (*Triticum aestivum*), especially their roles in the regulation of seed dormancy and germination. Here, we identified 73 members of the *IQD* gene family from the wheat genome and phylogenetically separated them into six major groups. Gene structure and conserved domain analyses suggested that most members of each group had similar structures. A chromosome positional analysis showed that *TaIQDs* were unevenly located on 18 wheat chromosomes. A synteny analysis indicated that segmental duplications played significant roles in *TaIQD* expansion, and that the *IQD* gene family underwent strong purifying selection during evolution. Furthermore, a large number of hormone, light, and abiotic stress response elements were discovered in the promoters of *TaIQDs*, implying their functional diversity. Microarray data for 50 *TaIQDs* showed different expression levels in 13 wheat tissues. Transcriptome data and a quantitative real-time PCR analysis of wheat varieties with contrasting seed dormancy and germination phenotypes further revealed that seven genes (*TaIQD4*/*-28*/*-32*/*-58*/*-64*/*-69*/*-71*) likely participated in seed dormancy and germination through the abscisic acid-signaling pathway. The study results provide valuable information for cloning and a functional investigation of candidate genes controlling wheat seed dormancy and germination; consequently, they increase our understanding of the complex regulatory networks affecting these two traits.

## 1. Introduction

Seed germination is a key event in the life cycles of all flowering plants that reproduce by seed. It includes several processes, from water absorption to radicle or embryonic protrusion [1]. Seed dormancy is a protective physiological characteristic that delays the timing of germination until the arrival of a favorable season [2]. For crops, dormancy has been gradually eliminated during breeding because it is not conducive to the rapid and uniform emergence of seedlings. Thus, pre-harvest sprouting (PHS, physiologically mature grains germinating in spikes before harvest) caused by dormancy loss occurs frequently in regions with prolonged wet weather during the harvest season, resulting in substantial losses in yield and quality [3]. The average annual loss caused by PHS in wheat (*Triticum aestivum*) worldwide exceeds USD 1 billion [4]. The PHS rate of rice exceeds 20% under the continuous high-temperature and rainy weather conditions in southern China [5]. Therefore, dormancy and germination are very important for crop production.

Wheat is the third largest cereal crop and is widely grown around the world. It is a staple food of human beings and also an important feed and industrial raw material [6,7]. However, wheat quality and yield are often affected by PHS owing to starch and protein degradation. Seed dormancy and germination are the main genetic factors that determine the PHS resistance level in wheat. Short-dormant wheat varieties are more prone to PHS than long-dormant varieties [8,9]. Therefore, the development of varieties with long periods of dormancy is essential for minimizing PHS damage in regions with prolonged rainy weather during the harvest season. Seed dormancy is jointly controlled by multiple genes; however, at present, only five genes, *TaVp1* [10], *TaMFT* (*TaPHS1*) [3,11], *TaSdr* [12,13], *TaMKK3-A* [14], and *TaQsd1* [15], underlying seed dormancy and PHS resistance have been cloned. Therefore, the identification of more novel genes associated with dormancy and germination will be helpful for breeding varieties with long periods of dormancy and high PHS resistance levels using the gene pyramiding method.

Calcium (Ca^2+^) is an important intracellular second messenger, and Ca^2+^-binding proteins act as Ca^2+^ sensors that enable plants to respond to biological processes by instantaneously regulating the Ca^2+^ concentration in the cytoplasm [16,17,18,19,20]. Ca^2+^ sensors in higher plants have been divided into four categories: calmodulins (CaMs), CaM-like proteins, calcineurin B-like proteins, and Ca^2+^-dependent protein kinases [21,22]. Among them, CaMs are important Ca^2+^ sensors in Ca^2+^ signal transduction [23]. They lack enzymatic activities, but they can bind and regulate the activities of CaM binding proteins [24,25]. The IQ67 domain (IQD) protein is a typical representative CaM binding protein that contains a central region of 67 conserved amino acid (aa) residues, named the IQ67 domain [26,27]. The IQ67 domain is precisely separated by three copies of the IQ motif, which are separated by short 11- and 15-aa sequences, respectively. In addition, each IQ pattern partially overlaps with the 1-8-14 and 1-5-10 motifs. The IQ67 domain contains three and four copies of the 1-8-14 and 1-5-10 motifs, respectively. In addition, several conserved hydrophobic and basic amino acid residues are present in these motifs [28,29,30]. These properties enable the IQ67 domain to form a basic amphiphilic helical structure, which establishes the specific functions of these proteins [31,32].

The *IQD* gene family members have been extensively identified in many plants, including 33 IQDs in Arabidopsis [31], 29 in rice [31], 26 in maize [30], 23 in Brachypodium [32], 29 in moso bamboo [33], 35 in Chinese cabbage [34], 49 in grapevine [35], 23 in potato [36], and 40 in poplar [37]. They can participate in developmental processes and responses to stress or hormonal changes. For example, in *Arabidopsis thaliana*, *AtIQD5*, *AtIQD14*, and *AtIQD16* participate in cell morphogenesis through Ca^2+^ signaling [27,38], and *AtIQD22* negatively regulates the accumulation of the plant hormone gibberellin [39]. *Brassica rapa* ssp. *pekinensis IQD5* plays a crucial role in responses to drought stress [34]. In *Zea mays*, the expression levels of 26 *ZmIQDs* have been shown to be transcriptionally regulated by a 20% polyethylene glycol (PEG) treatment [30]. Similarly, 12 *Populus trichocarpa*
*IQDs* also displayed responses to drought stress after 20% PEG treatment [37]. Notably, the tomato IQD-encoding gene *SUN24* positively regulates seed germination by repressing abscisic acid (ABA) signaling [40]. However, the roles of *IQD* genes in wheat seed dormancy and germination remain unclear.

In the present study, we used bioinformatics methods to identify *IQD* gene family members in the wheat genome. Additionally, detailed information regarding their phylogenetic relationships, gene structures, conserved motifs, synteny, evolutionary pattern, cis-acting elements, and the subcellular localizations were analyzed. Furthermore, the expression levels of the identified *TaIQDs* were investigated using a transcriptome analysis and quantitative real-time PCR (qRT-PCR) to validate their associations with seed dormancy and germination in wheat varieties havwithing contrasting dormancy and germination phenotypes. This study provides a theoretical basis for further dissecting the functions of the *IQD* genes in controlling wheat seed dormancy and germination.

## 2. Results

### 2.1. Identification of IQD Gene Family Members in Wheat

On the basis of the protein functional domain PF00612, redundant forms of the same gene were removed. A total of 73 IQD proteins were identified and renamed *TaIQD1*–*73* in accordance with their physical positions on the 21 wheat chromosomes. The detailed *TaIQD* gene characteristics, including their chromosomal positions, exon numbers, open reading frame (ORF) lengths, amino acid numbers, isoelectric points (pIs), molecular weights (MWs), instability index values, aliphatic indices (AIs), and grand averages of hydropathicity scores (GRAVYs), are summarized in Table 1. The ORF of the *TaIQD* genes ranged from 1020 bp (*TaIQD10*) to 1881 bp (*TaIQD38*), with predicted proteins of 339–626 aa. Correspondingly, the MWs of these TaIQD proteins ranged from 37,096.08 (*TaIQD10*) to 69,213.26 Da (*TaIQD30*). Although the derived TaIQD proteins showed diversity among some parameters, they were remarkably unified in terms of their relatively high pIs (pI > 9.38, with an average of 10.35), a value that is very similar to those of the IQD families in Chinese cabbage (10.05), *Brachypodium distachyon* (10.3), *A. thaliana* (10.3), and rice (10.4) [31,32,34]. Instability index calculations predicted that all the IQD proteins were unstable in vitro. The AI results showed that the thermal stabilities of the proteins ranged from 48.27 to 78.18, indicating that the differences in their thermal stability levels were relatively minor. The GRAVY scores for all the IQD proteins were negative, demonstrating that they were hydrophilic.

### 2.2. Phylogenetic Analysis and Conserved Sequence Alignment

To better clarify the evolutionary relationships between TaIQDs and the IQDs of other plant species, a phylogenetic tree was constructed by comparing identified AtIQD, ZmIQD, and OsIQD protein sequences (Figure 1a). Based on well-established maize IQD family classifications [30] and bootstrap values, the IQD proteins were classified into seven major groups. Among them, group III was the largest, containing 24 (48%) TaIQDs, 11 (22%) ZmIQDs, 8 (16%) OsIQDs, and 7 (14%) AtIQDs, and group VI formed the second largest clade, containing 17 (52%) TaIQDs, 6 (18%) ZmIQDs, 6 (18%) OsIQDs, and 4 (12%) AtIQDs (Figure 1b). The phylogenetic relationships indicated that the wheat IQD proteins were more homologous to IQDs from rice and maize than from Arabidopsis. The *IQD* gene identifiers from Arabidopsis, rice, and maize are listed in Appendix A.

To study the existence and locations of the TaIQD proteins’ conserved domains, we performed a multiple sequence alignment (Appendix A). We found that all the IQD family members contain a highly conserved IQD domain composed of 67 aa. The IQ67 domain was accurately separated by three copies of the IQ motif, IQxxxRGxxxR or (ILV)QxxxRxxxx(R, K), and the IQ motifs were separated by short sequences containing 11-aa residues and 15-aa residues, respectively. In addition, each IQ motif partially overlapped with three copies of the 1-8-14 motif ((FILVW) × 6(FAILVW) × 5(FILVW)) and four copies of the 1-5-10 motif ((FILVW) × 3(FILV) × 4(FILVW)). In addition to these motifs, both sides of each IQ motif were surrounded by hydrophobic and basic amino acid residues, which was consistent with a previous report [31].

### 2.3. Gene Structure and Motif Composition Analysis

To further investigate the evolutionary relationships among the wheat IQDs, we constructed a second phylogenetic tree using only the full-length TaIQD protein sequences. The IQD proteins were divided into six groups, I–VI, each having 3, 6, 29, 3, 15, and 17 members, respectively (Figure 2a). We identified 10 highly conserved motifs (motifs 1–10) in each IQD protein using MEME (Figure 2b, Appendix A). All the IQDs contained motif 1, and most of IQDs contained motifs 3 and 6. These constituted the most highly conserved part of the IQD domain. Interestingly, the motif compositions in the six groups were not exactly the same. For example, in Group III, motifs 1 and 3 existed in each member. However, *TaIQD4* and *TaIQD*28 both contained motifs 6, 7, and 8, whereas *TaIQD55* contained motifs 2, 4, and 5. These results showed the conservation and specificity of the gene structures and motif compositions of the *TaIQD* gene family members. The exon–intron organizational map analysis indicated that different numbers of exons (from 2 to 6) were found in the *TaIQDs* (Figure 2c). Interestingly, all the Group I members contained two exons, all the Group II members contained six exons, and all the Group IV members contained four exons. The number of introns in each wheat *IQD* gene ranged from one to five. In total, 21, 20, 14, 14, and 4 genes contained 2, 4, 3, 5, and 1 intron, respectively, indicating the structural diversity of the *TaIQDs*.

### 2.4. Analysis of the Chromosomal Locations and Duplications of TaIQD Genes

The 73 *TaIQDs* were unevenly distributed among the 21 wheat chromosomes, except for chromosomes 6A, 6B, and 6D (Figure 3a). Chromosomes 3A, 3B, and 3D contained the most *TaIQD* members (8, 10.96%), followed by chromosome 5A (7, 9.59%), chromosomes 1A, 1B, and 1D (6, 8.22%), and chromosome 5B (5, 6.85%). On the whole, 25, 25, and 23 *TaIQD* genes were detected in the A, B, and D sub-genomes, respectively, implying that no significant variation occurred in the IQD gene abundance on the sub-genome scale. Gene replication is an important mechanism that allows organisms to obtain new genes and create gene novelty [41]. Next, we evaluated the gene duplication events in the wheat *IQD* gene family. Tandem and segmental duplications are crucial for the evolution of gene families, allowing them to adapt to different environmental conditions. For the *TaIQD* genes, 111 gene pairs were detected as representing duplications (Figure 3a, Appendix A). Interestingly, no tandem duplication events among *IQD* genes were found. We further analyzed the Ka/Ks values of the duplicated gene pairs in wheat to understand the evolutionary constraints on *TaIQD* genes. The Ka/Ks ratios were always less than 1 (Figure 3b, Appendix A), indicating that the evolution of *TaIQD* genes was accompanied by intense purifying selection. The Ks values of the *TaIQD* duplicated gene pairs ranged from 0.0476 to 1.1249, with a concentration near 0.35; therefore, we speculated that the divergence time of the *TaIQD* gene pairs occurred approximately 26.92 million years ago.

### 2.5. Micro-Collinearity Analysis

To more thoroughly determine the phylogenetic mechanisms of *TaIQD* genes, we examined the synteny between wheat and four other gramineous species: *Ae. tauschii*, *Z. mays*, *T. dicoccoides*, and *O. sativa*. In total, 89, 116, 161, and 99 gene pairs were identified between hexaploid wheat and the other species, respectively (Figure 4a, Appendix A). Moreover, the average Ks values were 0.374 (Ta-Aet), 0.693 (Ta-Zm), 0.379 (Ta-Td), and 0.659 (Ta-Os), respectively (Figure 4b, Appendix A), indicating that the *TaIQD* gene family shared closer correlations with *A. tauschii* and *T. dicoccoides* than with rice and maize. In addition, the Ka/Ks ratios of all the collinear gene pairs among wheat and the four other gramineous plants were all less than 1, confirming that the evolution of the *IQD* gene family in wheat underwent a strong purifying selection.

### 2.6. Analysis of Cis Elements in TaIQD Promoters

To further explore the possible biological functions of the *TaIQD* genes, the presence of cis-acting elements in the 2 kb upstream promoter regions of the 73 *TaIQD* genes were predicted using the PlantCARE database (Figure 5, Appendix A). The identified cis-acting elements were mainly divided into three major categories: light, hormone, and abiotic stress-related elements. Light-related cis-elements included the TCCC-motif, I-box, GT1-motif, and G-box elements. Hormone-related cis-elements included the TCA-element (salicylic acid), ABA response-related element (ABRE), GARE motif (gibberellic acid), P-box element (gibberellic acid), TATC element (gibberellic acid), TGA element (auxin), AuxRR element (auxin), and TGACG motif (methyl jasmonate). Abiotic stress-related cis-elements included the drought-response (MYB binding site), low-temperature-response (CCGAAA), and the defense- and stress-responsive TC-rich elements. The most common elements in the *TaIQD* promoters were G-box, TGACG-motif, and ABRE, such as *TaIQD28*, *-58*, and *-64*, respectively. The presence of multiple cis-acting elements in the *TaIQD* promoters may be indicative of the encoded proteins’ diverse biological functions.

### 2.7. Expression Profiles of TaIQD Genes in Different Tissues

We obtained expression data for 50 *TaIQD* genes in 13 tissues related to wheat development from GEO accession number GSE12508 to analyze their expression profiles (Figure 6, Appendix A). Overall, most *TaIQDs* were widely expressed in many tissues. Interestingly, *TaIQD28*, *-32*, *-58*, *-64*, *-69*, and *-71* were highly expressed in germinated seeds and endosperm, suggesting that they are involved in the processes of seed dormancy and germination. However, *TaIQD56* and *TaIQD62* were only highly expressed in immature inflorescence and were very lowly expressed elsewhere.

### 2.8. Expression Pattern Analysis of TaIQDs in the Transcriptome

We also explored the expression patterns of *TaIQDs* from GEO accession number GSE49821 that has data from JM20 wheat seeds sampled at five successive phases, namely 0, 12, 24, 36, and 48 h after water imbibition, respectively (Figure 7a, Appendix A). The expression levels of 15 genes (*TaIQD24*/*-26*/*-30*/*-32*/*-34*/*-38*/*-39*/*-40*/*-42*/*-46*/*-58*/*-64*/*-67*/*-69*/*-71*) increased along with imbibition time, whereas those of 14 genes (*TaIQD6*/*-12*/*-18*/*-20*/*-27*/*-28*/*-35*/*-36*/*-43*/*-44*/*-48*/*-49*/*-51*/*-60*) decreased.

To further investigate the expression patterns of *TaIQD* genes during seed imbibition, two wheat varieties (HMC21 and J411) with contrasting dormancy phenotypes were sampled at different stages of seed imbibition (6, 9, and 12 h) for a transcriptome analysis (Figure 7b, Appendix A). Most of the TaIQD genes had significantly different expression levels in HMC21 and J411, except for *TaIQD70* and *TaIQD73*. Additionally, with the extension of seed imbibition time, more *TaIQD* genes became highly expressed. Particularly, the expression levels of 27 *TaIQD* genes (*TaIQD*/*-1*/*-2*/*-4*/*-8*/*-14*/*-21*/*-24*/*-26*/*-27*/*-32*/*-34*/*-39*/*-40*/*-41*/*-42*/*-43*/*-50*/*-52*/*-55*/*-57*/*-58*/*-61*/*-63*/*-64*/*-66*/*-69*/*-71*) in J411, which has a low dormancy level, were always higher than in HMC21, which has a high dormancy level, whereas eight *TaIQD* genes (*TaIQD7*/*-17*/*-23*/*-28*/*-36*/*-38*/*-53*/*-60*) exhibited the opposite pattern.

### 2.9. TaIQDs Expression Analysis by qRT-PCR

Combining the published JM20 transcriptome data and our own transcriptome data (HMC21 and J411), we identified 15 significantly differentially expressed genes, *TaIQD4*/*-24*/*-26*/*-28*/*-32*/*-34*/*-36*/*-39*/*-40*/*-42*/*-58*/*-60*/*-64*/*-69*/*-71*. To confirm the reliability of the two transcriptome data sets, we used qRT-PCR to further detect the expression levels of the above 15 *TaIQD* genes in six wheat varieties having contrasting dormancy levels after 0 h and 12 h of seed imbibition (Figure 8). After a 12 h imbibition, seeds from dormant varieties (HMC21, YXM, and YM16) showed no seed germination (average germination rate, 0%), whereas seeds from non-dormant varieties (J411, ZY9507, and ZM895) showed obvious seed germination (average germination rate, 98%). For each *TaIQD* gene, we also found obvious differences in relative transcript levels among the above six wheat varieties. In particular, *TaIQD4*/*-32*/*-58*/*-64*/*-69*/*-71* were more highly and consistently transcribed in the three varieties with low dormancy levels than in the three varieties with high dormancy levels. In contrast, *TaIQD28* revealed the opposite trend. Briefly, the qRT-PCR data supported the results of the above two transcriptomes.

The ABRE is a major cis-element found in ABA-responsive genes [42,43]. Owing to the presence of the ABRE element in the promoter regions of the above seven genes (*TaIQD4*/*-28*/*-32*/*-58*/*-64*/*-69*/*-71*) (Figure 5), we further analyzed their responses to exogenous ABA in HMC21 and J411 seeds to determine roles for *TaIQDs* in ABA-mediated germination. At 24 h after treatment (distilled water was the control treatment), all the HMC21 and J411 seeds treated with distilled water germinated. After the 50 μM ABA treatment, the germination behaviors of HMC21 and J411 seeds were inhibited, and the germination index values decreased to 53% and 82%, respectively, indicating that the sensitivity of seeds from the dormant variety HMC21 to ABA was greater than that of non-dormant J411, which was in accordance with our previous results [9]. In addition, we found that after the ABA treatment, the expression levels of *TaIQD4*/*-32*/*-58*/*-64*/*-69*/*-71* in HMC21 and J411 were significantly lower than after the distilled water treatment, whereas the expression level of *TaIQD28* was significantly higher (Figure 9), indicating that significant differences in ABA sensitivity exist among these seven *TaIQD* genes. Thus, the differences in ABA responses between dormant and non-dormant varieties may result from the differences among genes, in accordance with Walker-Simmons (1987).

### 2.10. Subcellular Localization Analysis

To assess the subcellular localizations of the TaIQDs, three fusion vectors were constructed and then transformed independently into rice protoplasts. As shown in Figure 10, TaIQD4-GFP was detected as being localized to cell membranes, whereas both TaIQD58-GFP and TaIQD64-GFP localized to nuclei and cell membranes. The GFP of the empty protein (35S-GFP), as the control group, was dispersed throughout the cell. Similar findings have also been reported for the IQD gene families of grapevine and Chinese cabbage [34,35].

## 3. Discussion

Plants have evolved many specific gene families to adapt to environmental changes. The *IQD* genes are a plant-specific family [35]. At present, the complete genome sequence of Chinese Spring wheat enables the comprehensive characterization of important gene families. In the current study, we identified 73 *TaIQD* genes from the wheat genome and found that there were more *TaIQD* members than in other plant species (such as *Arabidopsis*, *Zea mays*, and *O. sativa*), which may be attributed to the two rounds of polyploidization that occurred during wheat evolution [44]. Another possibility is that there are a large number of duplication events in the *TaIQDs*. Gene duplication events are of great significance for the rapid expansion and evolution of plant gene families [45].

Most genes in the same group shared similar gene structures in terms of intron number or exon length. Therefore, we speculated that the IQDs in one branch may have similar functions, which would be similar to the IQDs identified in other plants, such as *B. distachyon*, maize, and Chinese cabbage [30,32,34]. Furthermore, comparisons of the IQDs’ conserved structural domains revealed the complete IQ67 domain in all the *TaIQD* genes, suggesting that the IQ67 domain was highly conserved during evolution.

Phylogenetic analyses of plant *IQD* genes allow the study of their evolutionary history and the estimation of duplication events during the expansion of the *IQD* gene family. In this study, based on the full-length protein sequence of the TaIQDs, two monocotyledonous plants (maize and rice) and one dicotyledonous plant (Arabidopsis) were used to explore the phylogenetic relationship among IQDs. The 159 IQDs from the four plants were divided into seven groups (I–VII) through phylogenetic analysis. Interestingly, we found that even in different species, the number of IQD members was always the largest in Group III, followed by Group VI, and the number of IQD members was the least in Group I, implying the conservation of *IQD* genes during evolution. In addition, each group contained at least one gene from wheat, Arabidopsis, rice, and maize, indicating that members of different species may have evolved from the same ancestor.

The selection pressure on gene pairs (i.e., positive, purifying, and neutral) provides vital information related to divergence rate [46]. We found that all the duplicated gene pairs in the *TaIQD* gene family underwent strong purifying selection to remove harmful mutations at the protein level, which may contribute to maintaining their functional stability and explain the lack of divergence during evolution. The similar purifying selection of *IQD* genes in moso bamboo [33], Chinese cabbage [34], and grapevine [46] has been reported, indicating that the evolution of *TaIQD* genes was comparable with those of other plants. Additionally, segmental duplications might be the main driving force of *TaIQD* gene family expansion, consistent with a study of *IQD* genes in cotton [47].

Multiple cis-acting elements located in gene promoters play crucial roles in signaling [48]. In this study, we identified diverse cis-acting regulatory elements in the promoter regions of TaIQDs, including the G-box, ABRE, TGACG-motif, and MYB-binding site. In particular, the ABRE associated with ABA responsiveness was distributed widely throughout most of the *TaIQD* genes. ABA sensitivity (or response to ABA) is significantly correlated with seed dormancy and PHS resistance [49,50]. Thus, we speculated that the *TaIQDs* may be responsive to ABA and be involved in the ABA signaling pathway.

The functions of IQD family genes have also been widely studied. In potato, some *StIQDs* were expressed in a tissue-specific pattern, so they were closely related to tissue development [36]. In grapevine, 49 *VvIQDs* were related to the shape of grape berries [35]. *IQDs* were also found to be involved in abiotic stress response. For example, 12 IQD members were involved in poplar response to methyl jasmonate (MeJA) [37]. In bamboo, three pairs of duplicated genes (*Pe**IQD5*-*Pe**IQD26*, *P**e**IQD18*-*Pe**IQD21* and *Pe**IQD19*-*Pe**IQD23*) revealed different expression patterns under PEG treatment [33]. In our current study, after the ABA treatment, six genes (*TaIQD4*/*-32*/*-58*/*-64*/*-69*/*-71*) were down-regulated by ABA, whereas *TaIQD28* was up-regulated, implying that they may participate in seed dormancy and germination through the ABA signaling pathway. Our results provided basic information for further study of wheat IQD proteins as well as further enriched the functions of the *IQD* gene family. Bi et al. (2018) reported that overexpression of the tomato IQD-encoding gene *SUN24* promotes seed germination, whereas a knockdown of this gene delays germination. Further expression analyses showed that *SUN24* promotes seed germination by negatively regulating the expression levels of two key ABA signaling genes: *ABA-insensitive 3* and *5*. Moreover, the seed germination of the *SUN24* overexpression lines was less sensitive to ABA compared with the wild type. In contrast, the RNAi seeds of *SUN24* germinate slower than those of the wild type after ABA treatments. Strikingly, an evolutionary analysis showed that *TaIQD69* and *TaIQD71* were highly homologous to the tomato gene *SUN24*, supporting that these two genes were likely involved in regulating seed dormancy and germination through the ABA signaling pathway.

In addition to transcription levels, the molecular mechanisms of seed dormancy and germination have been shown at the DNA methylation level [51,52]. Interestingly, the DNA methylome data of ‘MingXian 169’ seeds showed that the methylation levels of the above five genes (*TaIQD4*/*-58*/*-64*/*-69*/*-71*) are significantly different between germinated and dormant seeds, with levels in dormant seeds tending to be higher than in germinated seeds [52]. These findings not only provide novel insights into the involvement of epigenetic modifications of *IQD* genes in regulating seed dormancy and germination, but also demonstrate the complex regulatory mechanisms of seed dormancy and germination through crosstalk among hormones and Ca^2+^ signaling pathways, as well as DNA methylation. We have learned that the characteristics of seed dormancy and germination determine the resistance of wheat to PHS: wheat varieties with a higher dormancy rate or lower germination rate have higher resistance to PHS. Therefore, mining the PHS resistance candidate genes of *TaIQD* and solving their regulation mechanism will help to reduce the loss of PHS to wheat yield and quality. In conclusion, our results provided valuable information for further cloning and functional analysis of *TaIQD* genes and candidate genes to improve PHS resistance in wheat.

## 4. Materials and Methods

### 4.1. Identification of IQD Genes in Wheat

Wheat genomic data were downloaded from Ensembl Plants (https://plants.ensembl.org/info/data/ftp/index.html/ accessed on 24 July 2021). The protein sequences of *A. thaliana IQD* family members were downloaded from the TAIR database (http://www.Arabidopsis.org/ accessed on 24 July 2021), and the published of IQD protein sequences from *Z. mays* and *Oryza sativa* were downloaded from the phytozome website (https://phytozome.jgi.doe.gov/pz/portal.html/ accessed on 24 July 2021). The IQD protein sequences from Arabidopsis, rice, and maize were used as queries to identify the putative TaIQD proteins in wheat through a local BLASTP program with a significant e-value (<le-5) [53]. After BLASTP, the PFAM database (http://pfam.xfam.org/search/ accessed on 24 July 2021) was used to further verify the presence of the conserved IQ domain PF00612 in the putative IQD protein sequences. Protein sequences with errors or lacking the domain were removed. The confirmed *IQD* genes were named in accordance with their positions on the wheat chromosomes. Protein physicochemical parameters, such as aliphatic indices (AIs), molecular weights (MWs), and grand averages of hydropathicity (GRAVY), were predicted using the ExPasy website (https://web.expasy.org/protparam/ accessed on 24 July 2021) [54].

### 4.2. Multiple Sequence Alignments and Phylogenetic Analysis

A multiple sequence alignment of all the identified TaIQD-conserved IQ67 domains was performed and shaded using DNAMAN (Version 6.0) software (http://www.lynnon.com/ accessed on 23 August 2021). The amino acid sequences of TaIQD were used to construct an unrooted phylogenetic tree by the neighbor-joining method in MEGA (Version 6.0) with 1000 bootstrap replicates [55,56]. To determine the relationships of IQDs among different species, their protein sequences from wheat, rice, Arabidopsis, and maize were used to construct a phylogenetic tree by the same methods described above.

### 4.3. Gene Structure, Conserved Motif, and Cis-Acting Element Analysis

To further understand the exon–intron structural features of *TaIQDs*, the coding sequences and their corresponding genomic sequences were analyzed using GSDS (http://gsds.gao-lab.org/ accessed on 23 August 2021) [57]. The conserved motifs of the identified *TaIQD* gene family members were predicted using the MEME online tool (https://meme-suite.org/meme/tools/meme/ accessed on 23 August 2021), with the following parameters: maximum number of 10 motifs and optimum motif widths of 6–50 residues. They were visualized using TBtools (Version 1.098) software [58]. On the basis of the wheat genome annotation, the 2000 bp regions upstream of the transcription start sites in all the verified *TaIQD* transcripts were extracted as promoters to predict the presence of cis-acting elements using Plant CARE with the default parameters (https://bioinformatics.psb.ugent.be/webtools/plantcare/html/ accessed on 23 August 2021) [59].

### 4.4. Chromosomal Location and Gene Duplication

The chromosome locational information for wheat genes was downloaded from the Ensembl Plants website (https://plants.ensembl.org/info/data/ftp/index.html/ accessed on 24 July 2021). All the *TaIQD* gene chromosomal locations were drafted and visualized in Chromosome-Basic Circos by TBtools. The Multiple Collinearity Scan toolkit (MCScanX) (Version 1.098) software was used to analyze gene duplication events within species [60], and the syntenic relationships of *IQD* genes with four other species (*Z. mays*, *O. sativa*, *Ae. tauschii*, and *T. dicoccoides*) were drawn using the Dual Systeny Plotter (Version 1.098) software in TBtools (Version 1.098) software (https://github.com/CJ-Chen/TBtools/ accessed on 24 July 2021) [61]. TBtools was used to calculate the non-synonymous substitution rate (Ka) to synonymous substitution rate (Ks) ratios [62]. Any Ks values > 2.0 were discarded owing to the risk of substitution saturation [30,63]. Ka/Ks values = 1, <1, and >1 represent neutral, negative (purifying), and positive selection, respectively [45]. Finally, the divergence time of collinear gene pairs (T) was determined as T = Ks/2λ × 10^−6^ million years ago, where λ = 6.5 × 10^−9^ [64].

### 4.5. Expression Pattern Analysis

The expression patterns of the identified *TaIQD* genes in 13 different tissues were analyzed using data from the GEO accession number GSE12508 on the National Center for Biotechnology Information (NCBI) website (https://www.ncbi.nlm.nih.gov/geo/query/acc.cgi?acc=GSE12508/ accessed on 24 July 2021). The published transcriptome data of ‘Jimai20’ (JM20) seeds after 0, 12, 24, 36, and 48 h of water absorption were obtained from the GEO accession number GSE49821 (https://www.ncbi.nlm.nih.gov/geo/query/acc.cgi?acc=GSE49821/ accessed on 24 July 2021).

To study the expression profiles of the 73 *TaIQD* genes, we collected seeds of ‘Hongmangchun21’ (HMC21) and ‘Jing 411’ (J411) subjected to water imbibition for the first time after 6, 9, and 12 h for transcriptome sequencing. *TaIQD* genes were obtained from the NetAffx Analysis Center (http://www.affymetrix.com/ accessed on 24 July 2021) to identify corresponding probe sets [9,65]. The FPKM values were log_2_ with (1+) conversion and displayed as TaIQD tissue specificity, and a heat map was used to reveal expression at different imbibition periods in seeds using TBtools (Version 1.098) software [61].

### 4.6. Plant Materials and Stress Treatments

Six wheat varieties with contrasting dormancy types were selected to investigate TaIQD expression patterns. Among them, the varieties with strong dormancy were HMC21, ‘Yangxiaomai’ (YXM), and ‘Yangmai16’ (YM16), and the weak dormancy varieties were J411, ‘Zhongyou9507’ (ZY9507), and ‘Zhongmai895’ (ZM895). Next, seeds of non-dormant HMC21 and J411 were treated with 50 μM ABA. Distilled water was used as a control treatment. Seeds were sampled at 24 h after treatment, immediately frozen in liquid nitrogen, and stored at −80 °C [9]. Three biological replications were performed.

### 4.7. RNA Isolation and qRT-PCR Analysis

Total RNA was isolated from frozen HMC21, YXM, YM16, J411, ZY9507, and ZM895 seed samples in accordance with the manufacturer’s instructions (AG, Changsha, China). The RNA was subsequently treated with DNaseI (AG, Changsha, China) to remove genomic DNA contamination. Upon RNA extraction, the quality and concentration of the RNA were detected by agarose gel electrophoresis. The RNA was then reverse transcribed into cDNA using a Primer Script RT reagent Kit (AG, Changsha, China). IQD gene-specific primers were designed using Primer software version 5.0 and used for the qRT-PCR analysis, with TaActin as the reference gene. qRT-PCR was conducted on a CFX96 Real-Time System (Bio-Rad, Hercules, CA, USA). Each reaction was conducted in a final volume of 20 µL, which included 10 µL of 2× SYBR Green Pro Taq HS Premix (AG, Changsha, China), 2.0 µL of diluted cDNA template, 0.4 µL of forward/reverse primer (10 μM), and 5.6 µL of ddH_2_O. The PCR parameters were programmed as follows; 95 °C for 30 s, 40 cycles at 95 °C for 5 s and 60 °C for 30 s, followed by a melting curve. The relative changes in gene expression were calculated using the 2^(−ΔΔCt)^ method [66]. Each sample was analyzed in three replicates. All the statistical analyses was carried out using GraphPad version 5 [67]. The significance criteria used were * *p* < 0.05 and ** *p* < 0.01.

### 4.8. Subcellular Localization Analysis

The coding sequences of *TaIQD4*, *TaIQD58*, and *TaIQD64* without the termination codons were cloned independently into the pCAMBIAI1305 vector containing the 35S-driven green fluorescent protein (GFP) sequence promoter. The membrane protein marker OsMCA1 was individually co-transferred into rice protoplasts with 35S-GFP (blank control) and TaIQD4-GFP. The TaIQD58-GFP and TaIQD64-GFP fusion proteins were co-transformed independently into rice protoplasts with the nuclear protein NLS. The transformed protoplasts were cultured in the dark at 22 °C. After 36 h, the GFP fluorescence signals were observed using a LSM710 confocal laser scanning microscope (CarlZeiss, Jena, Germany). The specific primers used, containing restriction sites *Xba*I and *BamH*I, are shown in Appendix A.

## 5. Conclusions

We identified 73 *TaIQD* genes in the wheat genome and subjected them to systematic bioinformatics analyses, including phylogenetic tree construction, gene structure, chromosomal gene distribution, promoter element, and gene collinearity. Subsequently, we performed transcriptome, qRT-PCR, and subcellular localization analyses. The seven genes *TaIQD4*/*-28*/*-32*/*-58*/*-64*/*-69*/*-71* were considered as candidate genes associated with seed dormancy and germination. These results will help to further determine the functions of *TaIQD* genes in regulating seed dormancy and germination, and thus, they have potential application values for the genetic improvement of wheat pre-harvest sprouting resistance.

## Figures and Tables

**Figure 1 ijms-23-04093-f001:**
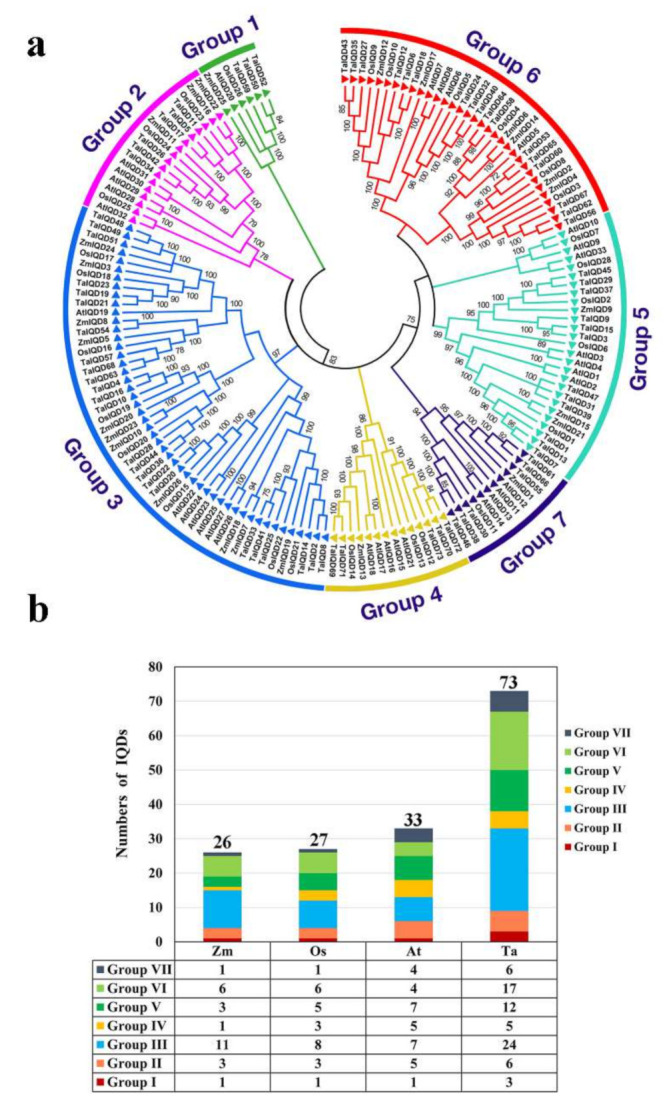
Phylogenetic analysis and distribution of IQ67 domain (IQD) proteins from wheat, Arabidopsis, rice, and maize. (**a**) The phylogenetic tree was constructed using the neighbor-joining method. The number of bootstrap values was 1000 replicates. (**b**) Statistics for *IQD* genes in each group from wheat, Arabidopsis, rice, and maize.

**Figure 2 ijms-23-04093-f002:**
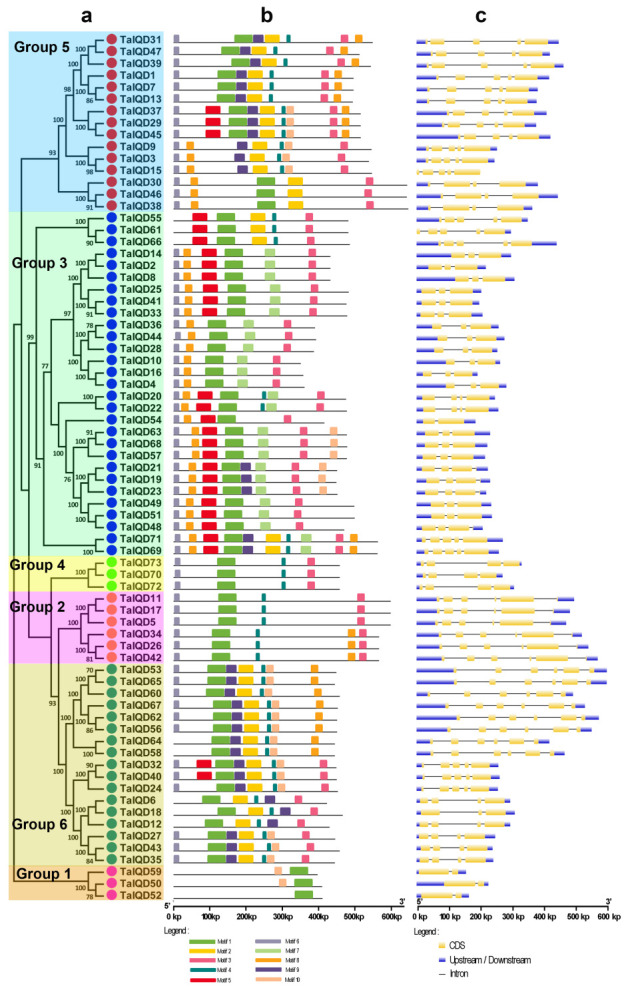
Phylogenetic relationships, gene structures, and conserved motifs among the 73 *TaIQDs*. (**a**) The phylogenetic tree was constructed with the full-length sequences of TaIQD proteins using MEGA6.0 software. (**b**) The motif compositions of the TaIQD proteins. Schematic representations of the 10 conserved motifs in the TaIQD proteins. The motifs, numbered 1–10, are indicated by different colored boxes. (**c**) Exon–intron structures of the *TaIQD* genes. Yellow boxes represent exons, grey lines indicate introns, and blue boxes represent untranslated 5′ and 3′ regions.

**Figure 3 ijms-23-04093-f003:**
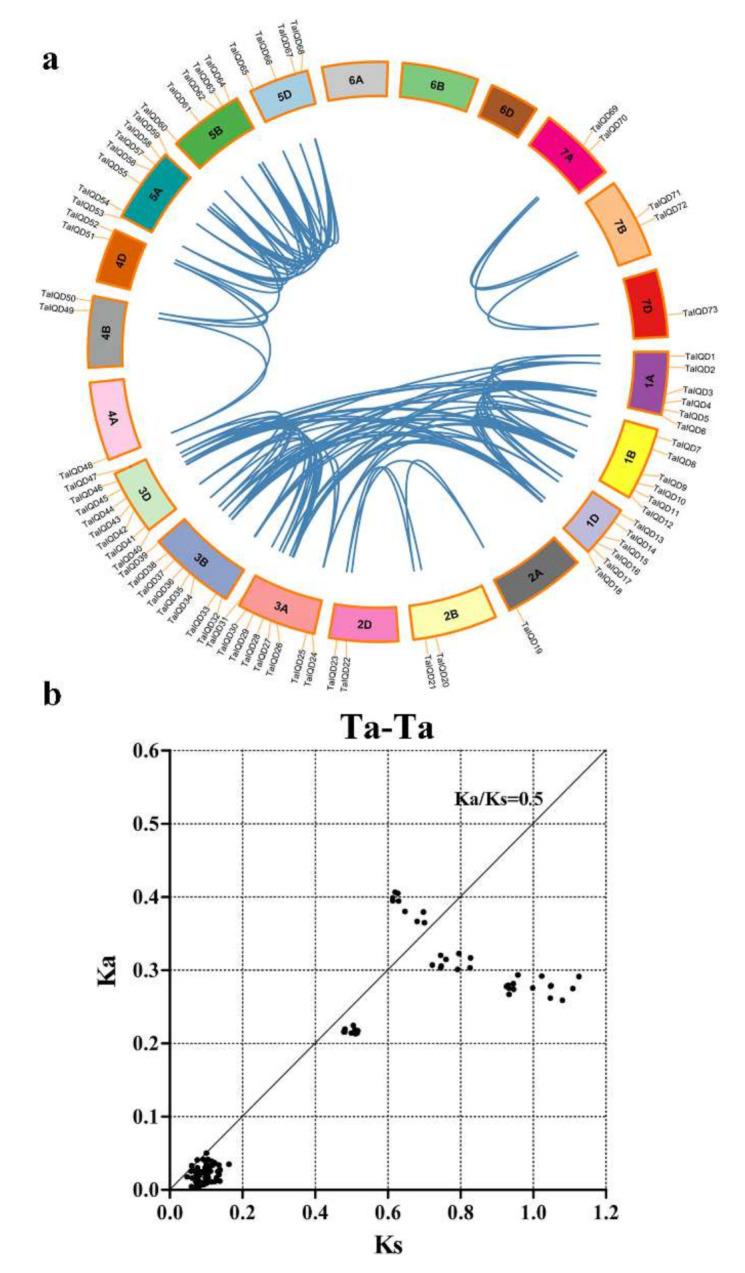
Chromosomal locations and gene duplications of *TaIQD* genes. (**a**) Chromosomal localizations of the *TaIQDs*. Different wheat chromosomes are presented in different colors. The blue lines indicate duplicated *IQD* gene pairs. The chromosome number is displayed next to each chromosome. (**b**) The Ka and Ks distributions of 111 duplicated gene pairs exhibited in a scatterplot.

**Figure 4 ijms-23-04093-f004:**
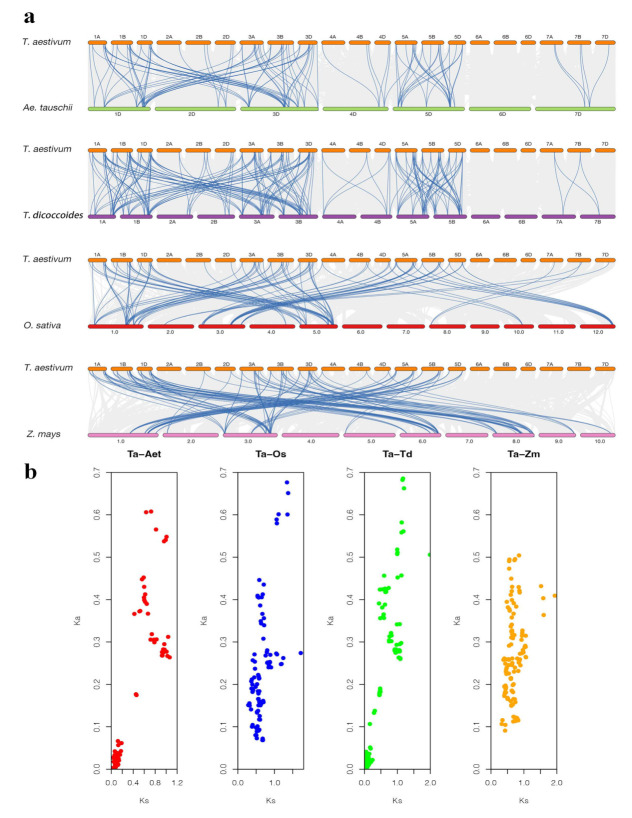
Synteny analysis of *TaIQD* genes between *T. aestivum* and four other plant species (*Zea mays*, *Oryza sativa*, *Aegilops tauschii*, and *Triticum dicoccoides*). (**a**) Grey lines in the background and blue lines between different species indicate the collinear blocks and syntenic IQD pairs between wheat and other species, respectively. (**b**) The scatter plot shows the Ka and Ks distributions of homologous pairs between wheat and the different species.

**Figure 5 ijms-23-04093-f005:**
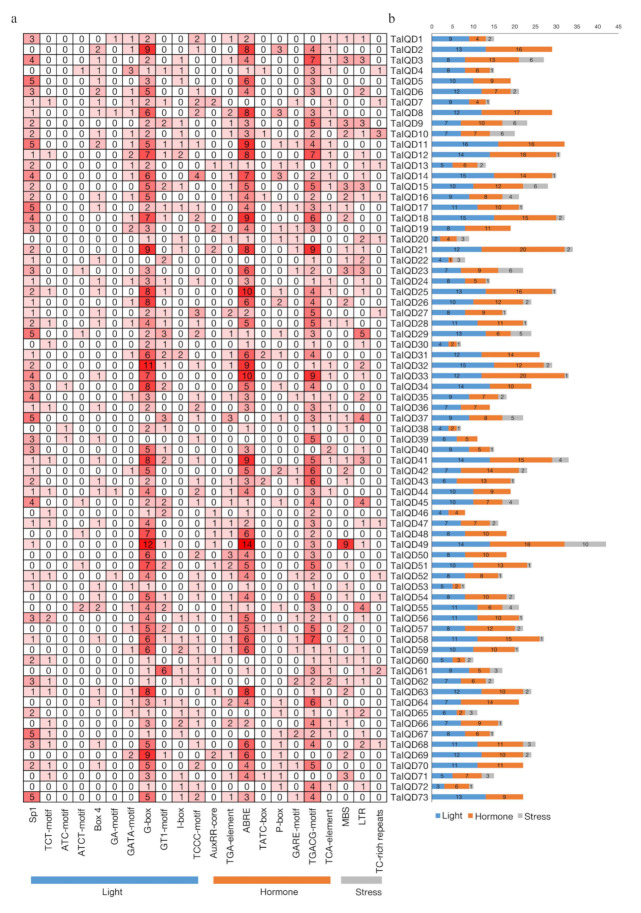
Analysis of cis-acting elements in the promoter regions of *TaIQD* genes. (**a**) The different colors and numbers of the grid indicated the numbers of different promoter elements in these *IQD* genes. (**b**) The different colored histogram represented the sum of the cis-acting elements in each category.

**Figure 6 ijms-23-04093-f006:**
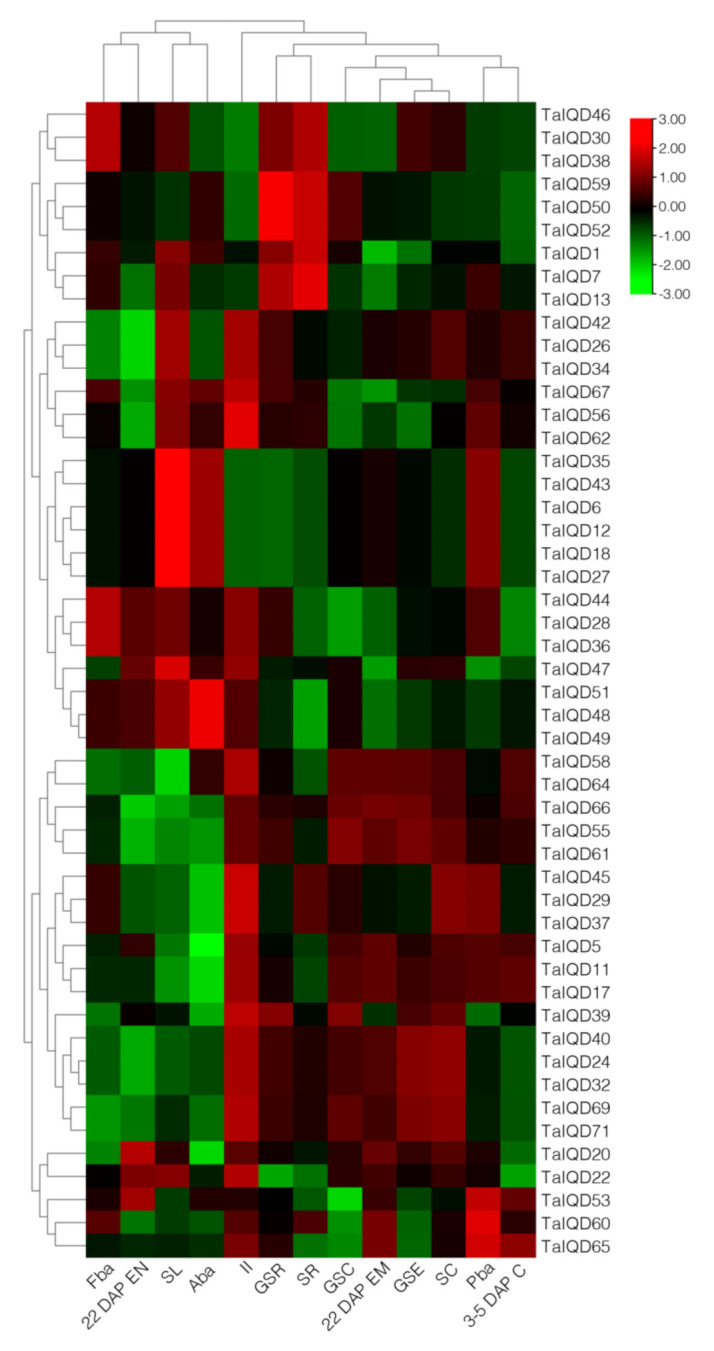
Expression profiles of *TaIQD* genes in different tissues and at different developmental stages. The heat map shows hierarchical clustering of the 50 *TaIQD* genes among different tissues. Abbreviations represent specific developmental stages: GSC, germinating seed, coleoptile; GSR, germinating seed, root; GSE, germinating seed, embryo; SR, seedling, root; SC, seedling, crown; SL, seedling, leaf; II, immature inflorescence; FBA, floral bracts, before anthesis; PBA, pistil, before anthesis; Aba, anthers, before anthesis; 3–5 DAP C, 3–5 DAP caryopsis; 22 DAP EM, 22 DAP embryo; 22 DAP EN, 22 DAP endosperm.

**Figure 7 ijms-23-04093-f007:**
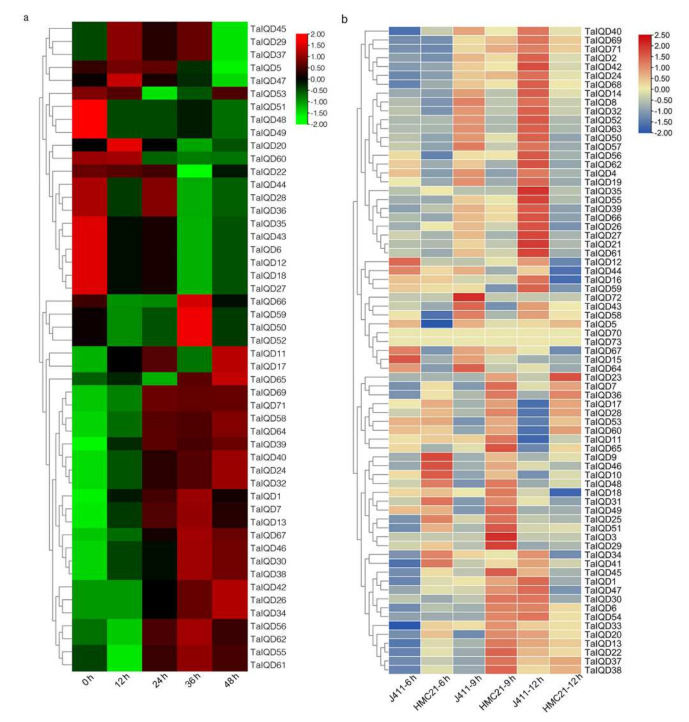
Heat map showing the hierarchical clustering of *TaIQDs* from different databases. (**a**) GSE49821. Abbreviations represent ‘Jimai20’ (JM20) germinating seed imbibition times: 0, 12, 24, 36, and 48 h. (**b**) Imbibition times of ‘Hongmangchun21’ (HMC21)- and ‘Jing411’ (J411)-germinated seeds: 6, 9, and 12 h.

**Figure 8 ijms-23-04093-f008:**
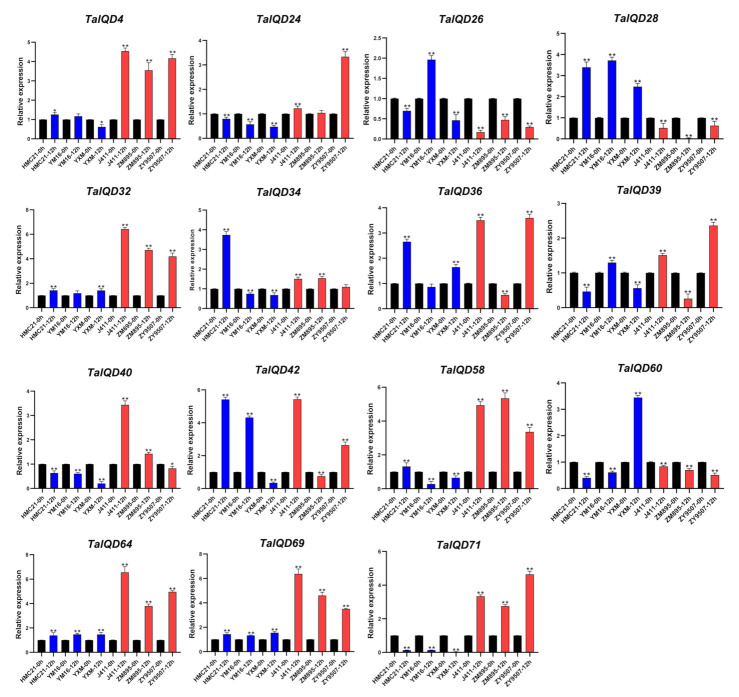
Expression patterns of 15 *TaIQDs* during seed imbibition in six wheat varieties with contrasting seed dormancy phenotypes. Red indicates weak-dormancy varieties, and blue indicates strong-dormancy variety. Y axes represent the scales of the relative expression levels. Bars represent the standard deviations (SDs) of three biological replicates. The significance criteria used were * *p* < 0.05 and ** *p* < 0.01.

**Figure 9 ijms-23-04093-f009:**
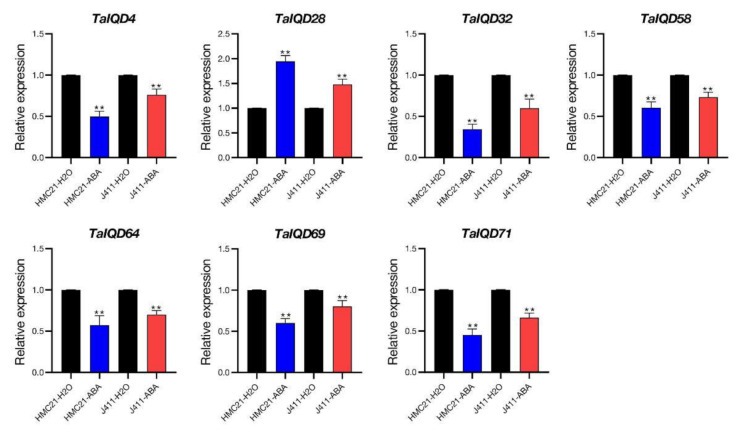
Expression patterns of seven *TaIQD* genes after 50-μM ABA treatment in wheat varieties ‘Hongmangchun21’ (HMC21) and ‘Jing411’ (J411). Red indicates weak-dormancy varieties, and blue indicates strong-dormancy varieties. Y axes represent the scales of the relative expression levels. Bars represent the standard deviations (SDs) of three biological replicates. The significance criteria used were ** *p* < 0.01.

**Figure 10 ijms-23-04093-f010:**
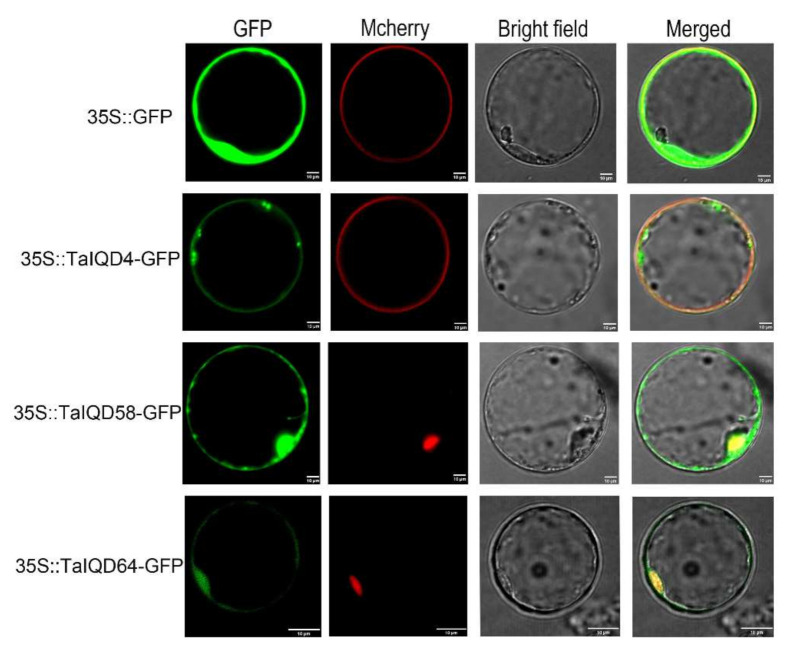
Subcellular localizations of TaIQD4, TaIQD58, and TaIQD64 in rice protoplasts. Scale bars = 10 μm. 35S::GFP was used as the empty control.

**Table 1 ijms-23-04093-t001:** Detailed information of the 73 predicted IQ67 domain (IQD) proteins in *Triticum aestivum*.

Name	Gene ID	Location	Open Reading Frame Length (bp)	Size (Amino Acid)	Molecular Weight (Da)	Isoelectric Points	Exons	Instability Index	Aliphatic Indices	Grand Averages of Hydropathicity
TaIQD1	TraesCS1A02G064100	Chr1A:45793606-45797826	1443	480	52,361.54	10.15	5	Unstable	71.17	−0.71
TaIQD2	TraesCS1A02G122900	Chr1A:140960645-14096280	1257	418	44,891.95	10.46	3	Unstable	65.67	−0.47
TaIQD3	TraesCS1A02G288100	Chr1A:485337870-48534035	1566	521	56,328.25	10.55	5	Unstable	48.27	−0.938
TaIQD4	TraesCS1A02G341900	Chr1A:531281089-53128394	1050	349	37,937.17	10.66	3	Unstable	65.56	−0.672
TaIQD5	TraesCS1A02G357500	Chr1A:540047888-54005224	1740	579	64,209.32	9.75	6	Unstable	66.94	−0.816
TaIQD6	TraesCS1A02G362400	Chr1A:543060516-54306349	1230	409	44,715.09	10.71	4	Unstable	61.69	−0.721
TaIQD7	TraesCS1B02G082100	Chr1B:65547252-65551108	1443	480	52,439.53	10.07	5	Unstable	68.71	−0.746
TaIQD8	TraesCS1B02G142100	Chr1B:190629188-19063230	1257	418	44,925.92	10.46	3	Unstable	64.5	−0.488
TaIQD9	TraesCS1B02G297500	Chr1B:517901878-51790443	1587	528	56,591.61	10.63	5	Unstable	52.27	−0.863
TaIQD10	TraesCS1B02G354600	Chr1B:584324452-58432711	1020	339	37,096.08	10.95	3	Unstable	64.07	−0.755
TaIQD11	TraesCS1B02G374100	Chr1B:604548650-60455366	1740	579	63,871.93	9.56	6	Unstable	68	−0.778
TaIQD12	TraesCS1B02G379600	Chr1B:612863317-61286629	1251	416	45,415.84	10.66	4	Unstable	61.35	−0.736
TaIQD13	TraesCS1D02G064700	Chr1D:46181934-46185754	1437	478	52,189.3	10.07	5	Unstable	69.44	−0.726
TaIQD14	TraesCS1D02G123800	Chr1D:126471856-12647475	1257	418	44,775.75	10.46	3	Unstable	64.98	−0.484
TaIQD15	TraesCS1D02G287100	Chr1D:385615809-38561783	1590	529	56,884.84	10.54	5	Unstable	51.1	−0.914
TaIQD16	TraesCS1D02G344200	Chr1D:432807534-43280947	1041	346	37,654.85	10.88	3	Unstable	64.45	−0.703
TaIQD17	TraesCS1D02G361800	Chr1D:444361818-44436669	1740	579	64,058.13	9.69	6	Unstable	67.13	−0.805
TaIQD18	TraesCS1D02G367300	Chr1D:447254643-44725776	1356	451	49,366.40	10.42	3	Unstable	65.23	−0.623
TaIQD19	TraesCS2A02G470100	Chr2A:712980746-71298308	1305	434	47,480.54	10.08	4	Unstable	61.59	−0.658
TaIQD20	TraesCS2B02G418900	Chr2B:600440674-60044316	1383	460	49,548.51	10.55	4	Unstable	51.78	−0.78
TaIQD21	TraesCS2B02G492900	Chr2B:690907263-69090953	1311	436	47,526.50	10.20	4	Unstable	58.85	−0.663
TaIQD22	TraesCS2D02G398300	Chr2D:511437009-51143961	1389	462	49,645.65	10.56	4	Unstable	53.7	−0.747
TaIQD23	TraesCS2D02G470000	Chr2D:574631280-57463349	1314	437	47,638.78	10.22	4	Unstable	60.07	−0.655
TaIQD24	TraesCS3A02G105600	Chr3A:69504313-69506901	1317	438	47,911.68	10.56	5	Unstable	70.75	−0.609
TaIQD25	TraesCS3A02G108100	Chr3A:73512378-73514437	1404	467	49,931.39	10.16	3	Unstable	64.84	−0.518
TaIQD26	TraesCS3A02G244100	Chr3A:457460875-45746634	1647	548	60,598.78	9.78	6	Unstable	75.88	−0.694
TaIQD27	TraesCS3A02G251400	Chr3A:471278391-47128089	1296	431	47,074.55	10.71	4	Unstable	62.09	−0.741
TaIQD28	TraesCS3A02G275900	Chr3A:505523600-50552617	1125	374	41,766.59	10.81	3	Unstable	61.5	−0.794
TaIQD29	TraesCS3A02G352500	Chr3A:600339974-60034378	1500	499	55,768.28	10.27	5	Unstable	61.86	−0.833
TaIQD30	TraesCS3A02G398900	Chr3A:645637395-64564125	1872	623	69,213.26	11.35	4	Unstable	50.14	−0.906
TaIQD31	TraesCS3A02G538100	Chr3A:749341062-74934558	1596	531	56,655.68	10.23	5	Unstable	64.29	−0.645
TaIQD32	TraesCS3B02G124000	Chr3B:97076190-97078793	1305	434	47,645.28	10.56	5	Unstable	70.05	−0.65
TaIQD33	TraesCS3B02G127100	Chr3B:104970381-10497248	1392	463	49,945.46	10.17	3	Unstable	64.73	−0.541
TaIQD34	TraesCS3B02G276000	Chr3B:445602528-44560775	1647	548	60,603.81	9.85	6	Unstable	78.18	−0.692
TaIQD35	TraesCS3B02G280900	Chr3B:451530157-45153259	1293	430	46,916.43	10.65	4	Unstable	63.37	−0.718
TaIQD36	TraesCS3B02G309600	Chr3B:498325074-49832768	1134	377	42,099.99	11.00	3	Unstable	63.58	−0.78
TaIQD37	TraesCS3B02G385100	Chr3B:605267059-60527119	1500	499	55,811.31	10.27	5	Unstable	61.86	−0.832
TaIQD38	TraesCS3B02G431200	Chr3B:670023177-67002686	1881	626	69,491.65	11.41	4	Unstable	49.44	−0.907
TaIQD39	TraesCS3B02G603500	Chr3B:822903185-82290786	1581	526	56,241.10	10.31	5	Unstable	64.35	−0.671
TaIQD40	TraesCS3D02G107700	Chr3D:61311367-61314011	1308	435	47,677.38	10.56	5	Unstable	69.89	−0.632
TaIQD41	TraesCS3D02G109900	Chr3D:63778901-63780895	1386	461	49,561.05	10.24	3	Unstable	65.86	−0.508
TaIQD42	TraesCS3D02G247300	Chr3D:346354612-34636037	1647	548	60,756.97	9.84	6	Unstable	75.33	−0.734
TaIQD43	TraesCS3D02G251800	Chr3D:352612821-35261524	1332	443	48,523.32	10.68	4	Unstable	63.27	−0.728
TaIQD44	TraesCS3D02G275900	Chr3D:382629319-38263211	1143	380	42,344.28	10.94	3	Unstable	63.87	−0.766
TaIQD45	TraesCS3D02G346500	Chr3D:457554464-45755872	1500	499	55,768.28	10.27	5	Unstable	62.06	−0.829
TaIQD46	TraesCS3D02G392900	Chr3D:508079513-50808400	1869	622	69,151.24	11.35	4	Unstable	50.39	−0.899
TaIQD47	TraesCS3D02G543500	Chr3D:613703281-61370752	1491	496	53,101.63	10.14	5	Unstable	61.96	−0.702
TaIQD48	TraesCS4A02G028700	Chr4A:21051778-21053883	1368	455	48,711.53	11.15	4	Unstable	58.31	−0.647
TaIQD49	TraesCS4B02G277100	Chr4B:558725978-55872825	1449	482	52,255.78	11.16	3	Unstable	63.94	−0.547
TaIQD50	TraesCS4B02G330700	Chr4B:621671148-62167343	1191	396	40,618.63	11.33	2	Unstable	72.5	−0.2
TaIQD51	TraesCS4D02G275700	Chr4D:446651522-44665391	1452	483	52,450.89	11.24	3	Unstable	62.57	−0.606
TaIQD52	TraesCS4D02G327500	Chr4D:486907798-48690946	1194	397	40,630.77	11.47	2	Unstable	70.88	−0.206
TaIQD53	TraesCS5A02G029600	Chr5A:25698699-25704674	1305	434	46,543.00	10.25	6	Unstable	60.88	−0.961
TaIQD54	TraesCS5A02G055100	Chr5A:51736382-51738258	1209	402	42,506.48	10.86	2	Unstable	58.81	−0.65
TaIQD55	TraesCS5A02G163000	Chr5A:348717683-34872122	1401	466	51,850.83	10.25	5	Unstable	65.9	−0.7
TaIQD56	TraesCS5A02G373000	Chr5A:571149994-57115556	1311	436	47,752.33	9.69	6	Unstable	67.43	−0.869
TaIQD57	TraesCS5A02G377500	Chr5A:574931819-57493400	1389	462	49,373.67	10.33	3	Unstable	62.97	−0.513
TaIQD58	TraesCS5A02G425800	Chr5A:611132861-61113757	1293	430	47,774.71	10.00	6	Unstable	68.81	−0.805
TaIQD59	TraesCS5A02G502100	Chr5A:667123853-66712542	1155	384	39,565.30	11.39	2	Unstable	71.69	−0.267
TaIQD60	TraesCS5B02G028200	Chr5B:27294672-27299651	1332	443	47,698.23	10.02	6	Unstable	60.47	−0.953
TaIQD61	TraesCS5B02G160400	Chr5B:295380499-29538350	1401	466	50,900.00	10.72	5	Unstable	64.87	−0.649
TaIQD62	TraesCS5B02G375000	Chr5B:552482573-55248837	1311	436	48,004.64	9.64	6	Unstable	68.33	−0.855
TaIQD63	TraesCS5B02G381100	Chr5B:558870075-55887241	1389	462	49,424.76	10.44	3	Unstable	62.34	−0.525
TaIQD64	TraesCS5B02G427800	Chr5B:603883827-60388805	1293	430	47,671.63	9.89	6	Unstable	70.44	−0.817
TaIQD65	TraesCS5D02G037300	Chr5D:36476089-36482142	1293	430	46,336.84	10.17	6	Unstable	62.56	−0.955
TaIQD66	TraesCS5D02G168000	Chr5D:262838385-26284284	1413	470	52,167.06	10.16	5	Unstable	65.74	−0.704
TaIQD67	TraesCS5D02G382500	Chr5D:452235938-45224129	1317	438	48,184.85	9.72	6	Unstable	65.8	−0.896
TaIQD68	TraesCS5D02G387500	Chr5D:457006913-45700916	1392	463	49,554.10	10.47	3	Unstable	63.05	−0.506
TaIQD69	TraesCS7A02G317400	Chr7A:457568641-45757125	1635	544	59,224.93	10.73	5	Unstable	54.56	−0.787
TaIQD70	TraesCS7A02G332500	Chr7A:485743582-48574631	1332	443	48,287.92	10.41	4	Unstable	58.8	−0.729
TaIQD71	TraesCS7B02G218800	Chr7B:408567032-40856977	1638	545	59,385.10	10.78	5	Unstable	54.46	−0.799
TaIQD72	TraesCS7B02G244900	Chr7B:453334625-45333772	1332	443	48,516.26	10.4	4	Unstable	59.23	−0.757
TaIQD73	TraesCS7D02G341000	Chr7D:436877993-43688133	1332	443	48,511.23	10.55	4	Unstable	58.13	−0.774

## Data Availability

Not applicable.

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
