# Peer review of "Identification of the Wheat (Triticum aestivum) IQD Gene Family and an Expression Analysis of Candidate Genes Associated with Seed Dormancy and Germination"

_ijms, 2022, doi:10.3390/ijms23084093_

Round 1

Reviewer 1 Report

The manuscript Identification of the wheat (Triticum aestivum) IQD gene family and an expression analysis of candidate genes associated with seed dormancy and germination is interesting and well written.

The abstract section is OK.

-  Introduction section is well prepared and well supported with up-to-date literature but in Line 82, Zea mays must be italic.

- Results section is properly described, data are really well presented but in Line 104 write full name before MWs.

- Discussion section is OK.

- In Table 1, write the full name for each abbreviation such as IQD, MW, ORF....etc

- In Figure 1, write the full name for IQD.

- In Figure 8, the X-axis needs to increase the resolution or must be bold.

- Materials and Methods section is OK.

- Conclusion section is OK.

Reviewer 2 Report

In the summary, the results obtained should be compared and discussed more intensively with data from other publications.

The use of the current results for later application at the scientific level as well as on an applied basis (e.g. plant breeding) needs to be elaborated in more detail.
